# Self-attention with Functional Time Representation Learning

**Da Xu**[*], **Chuanwei Ruan**[*], **Sushant Kumar** , **Evren Korpeoglu** , **Kannan Achan**
Walmart Labs
California, CA 94086
{Da.Xu,Chuanwei.Ruan,EKorpeoglu,SKumar4,KAchan}@walmartlabs.com

## Abstract

Sequential modelling with self-attention has achieved cutting edge performances in natural language processing. With advantages in model flexibility, computation complexity and interpretability, self-attention is gradually becoming a key component in event sequence models. However, like most other sequence models, self-attention does not account for the time span between events and thus captures sequential signals rather than temporal patterns. Without relying on recurrent network structures, self-attention recognizes event orderings via positional encoding. To bridge the gap between modelling time-independent and time-dependent event sequence, we introduce a functional feature map that embeds time span into high-dimensional spaces. By constructing the associated translation-invariant time kernel function, we reveal the functional forms of the feature map under classic functional function analysis results, namely Bochner's Theorem and Mercer's Theorem. We propose several models to learn the functional time representation and the interactions with event representation. These methods are evaluated on real-world datasets under various continuous-time event sequence prediction tasks. The experiments reveal that the proposed methods compare favorably to baseline models while also capturing useful time-event interactions.

## 1   Introduction

Attention mechanism, which assumes that the output of an event sequence is relevant to only part of the sequential input, is fast becoming an essential instrument for various machine learning tasks such as neural translation [1], image caption generation [25] and speech recognition [4]. It works by capturing the importance weights of the sequential inputs successively and is often used as an add-on component to base models such as recurrent neural networks (RNNs) and convolutional neural networks (CNNs) [3]. Recently, a seq-to-seq model that relies only on an attention module called 'self-attention' achieved state-of-the-art performance in neural translation [20]. It detects attention weights from input event sequence and returns the sequence representation. Without relying on recurrent network structures, self-attention offers appealing computational advantage since sequence processing can be fully parallelized. Key to the original self-attention module is positional encoding, which maps discrete position index $\{1, \dots, l\}$ to a vector in $\mathbb{R}^d$ and can be either fixed or jointly optimized as free parameters. Positional encoding allows self-attention to recognize ordering information. However, it also restricts the model to time-independent or discrete-time event sequence modelling where the difference in ordered positions can measure distance between event occurrences.

In continuous-time event sequences, the time span between events often has significant implications on their relative importance for prediction. Since the events take place aperiodically, there are gaps between the sequential patterns and temporal patterns. For example, in user online behaviour analysis,

---

[*]The two first authors contribute equally to this work.

the dwelling time often indicates the degree of interest on the web page while sequential information considers only the ordering of past browsing. Also, detecting interactions between temporal and event contexts is an increasingly important topic in user behavioural modelling [12]. In online shopping, transactions usually indicate long-term interests, while views are often short-termed. Therefore future recommendations should depend on both event contexts and the timestamp of event occurrences.

To effectively encode the event contexts and feed them to self-attention models, the discrete events are often embedded into a continuous vector space [2]. After training, the inner product of their vector representations often reflect relationship such as similarity. In ordinary self-attention, the event embeddings are often added to positional encoding to form an event-position representation [20]. Therefore, it is natural and straightforward to think about replacing positional encoding with some functional mapping that embeds time into vector spaces.

However, unlike positional encoding where representations are needed for only a finite number of indices, time span is a continuous variable. The challenges of embedding time are three folds. Firstly, a suitable functional form that takes time span as input needs to be identified. Secondly, the functional form must be properly parameterized and can be jointly optimized as part of the model. Finally, the embedded time representation should respect the function properties of time itself. To be specific, relative time difference plays far more critical roles than absolute timestamps, for both interpolation or extrapolation purposes in sequence modelling. Therefore, the relative positions of two time representations in the embedding space should be able to reflect their temporal difference. The contributions of our paper are concluded below:

- We propose the translation-invariant *time kernel* which motivates several functional forms of *time feature mapping* justified from classic functional analysis theories, namely Bochner's Theorem [13] and Mercer's Theorem [15]. Compared with the other heuristic-driven time to vector methods, our proposals come with solid theoretical justifications and guarantees.

- We develop feasible *time embeddings* according to the *time feature mappings* such that they are properly parameterized and compatible with self-attention. We further discuss the interpretations of the proposed *time embeddings* and how to model their interactions with event representations under self-attention.

- We evaluate the proposed methods qualitatively and quantitatively and compare them with several baseline methods in various event prediction tasks with several datasets (two are public). We specifically compare with RNNs and self-attention with positional encoding to demonstrate the superiority of the proposed approach for continuous-time event sequence modelling. Several case studies are provided to show the time-event interactions captured by our model.

## 2  Related Work

The original self-attention uses dot-product attention [20], defined via:

$$\text{Attn}(\mathbf{Q}, \mathbf{K}, \mathbf{V}) = \text{softmax}\Big(\frac{\mathbf{Q}\mathbf{K}^\top}{\sqrt{d}}\Big)\mathbf{V}, \tag{1}$$

where $\mathbf{Q}$ denotes the queries, $\mathbf{K}$ denotes the keys and $\mathbf{V}$ denotes the values (representations) of events in the sequence. Self-attention mechanism relies on the *positional encoding* to recognize and capture sequential information, where the vector representation for each position, which is shared across all sequences, is added or concatenated to the corresponding event embeddings. The above $\mathbf{Q}$, $\mathbf{K}$ and $\mathbf{V}$ matrices are often linear (or identity) projections of the combined event-position representations. Attention patterns are detected through the inner products of query-key pairs, and propagate to the output as the weights for combining event values. Several variants of self-attention have been developed under different use cases including online recommendation [10], where sequence representations are often given by the attention-weighted sum of event embeddings.

To deal with continuous time input in RNNs, a time-LSTM model was proposed with modified gate structures [27]. Classic temporal point process also allows the usage of inter-event time interval as continuous random variable in modelling sequential observations [26]. Several methods are proposed to couple point process with RNNs to take account of temporal information [23, 22, 14, 6]. In these work, however, inter-event time intervals are directly appended to hidden event representations as inputs to RNNs. A recent work proposes a time-aware RNN with time encoding [12].

The functional time embeddings proposed in our work have sound theoretical justifications and interpretations. Also, by replacing positional encoding with time embedding we inherit the advantages of self-attention such as computation efficiency and model interpretability. Although in this paper we do not discuss how to adapt the function time representation to other settings, the proposed approach can be viewed as a general time embedding technique.

## 3 Preliminaries

Embedding time from an interval (suppose starting from origin) $T = [0, t_{\max}]$ to $\mathbb{R}^d$ is equivalent to finding a mapping $\Phi : T \to \mathbb{R}^d$. Time embeddings can be added or concatenated to event embedding $Z \in \mathbb{R}^{d_E}$, where $Z_i$ gives the vector representation of event $e_i$, $i = 1, \dots, V$ for a total of $V$ events. The intuition is that upon concatenation of the event and time representations, the dot product between two time-dependent events $(e_1, t_1)$ and $(e_2, t_2)$ becomes $\left[Z_1, \Phi(t_1)\right]^{'}\left[Z_2, \Phi(t_2)\right] = \langle Z_1, Z_2 \rangle + \langle \Phi(t_1), \Phi(t_2) \rangle$. Since $\langle Z_1, Z_2 \rangle$ represents relationship between events, we expect that $\langle \Phi(t_1), \Phi(t_2) \rangle$ captures temporal patterns, specially those related with the temporal difference $t_1 - t_2$ as we discussed before. This suggests formulating temporal patterns with a translation-invariant kernel $\mathcal{K}$ with $\Phi$ as the *feature map* associated with $\mathcal{K}$.

Let the kernel be $\mathcal{K} : T \times T \to \mathbb{R}$ where $\mathcal{K}(t_1, t_2) := \langle \Phi(t_1), \Phi(t_2) \rangle$ and $\mathcal{K}(t_1, t_2) = \psi(t_1 - t_2), \forall t_1, t_2 \in T$ for some $\psi : [-t_{\max}, t_{\max}] \to \mathbb{R}$. Here the feature map $\Phi$ captures how kernel function embeds the original data into a higher dimensional space, so the idea of introducing the time kernel function is in accordance with our original goal. Notice that the kernel function $\mathcal{K}$ is positive semidefinite (PSD) since we have expressed it with a Gram matrix. Without loss of generality we assume that $\Phi$ is continuous, which indicates that $\mathcal{K}$ is translation-invariant, PSD and also continuous.

So the task of learning temporal patterns is converted to a kernel learning problem with $\Phi$ as feature map. Also, the interactions between event embedding and time can now be recognized with some other mappings as $\big(Z, \Phi(t)\big) \mapsto f\big(Z, \Phi(t)\big)$, which we will discuss in Section 6. By relating time embedding to kernel function learning, we hope to identify $\Phi$ with some functional forms which are compatible with current deep learning frameworks, such that computation via bask-propagation is still feasible. Classic functional analysis theories provides key insights for identifying candidate functional forms of $\Phi$. We first state Bochner's Theorem and Mercer's Theorem and briefly discuss their implications.

**Theorem 1** (Bochner's Theorem). *A continuous, translation-invariant kernel $\mathcal{K}(\mathbf{x}, \mathbf{y}) = \psi(\mathbf{x} - \mathbf{y})$ on $\mathbb{R}^d$ is positive definite if and only if there exists a non-negative measure on $\mathbb{R}$ such that $\psi$ is the Fourier transform of the measure.*

The implication of Bochner's Theorem is that when scaled properly we can express $\mathcal{K}$ with:

$$\mathcal{K}(t_1, t_2) = \psi(t_1, t_2) = \int_{\mathbb{R}} e^{i\omega(t_1 - t_2)} p(\omega) d\omega = E_\omega\big[\xi_\omega(t_1)\xi_\omega(t_2)^*\big], \tag{2}$$

where $\xi_\omega(t) = e^{i\omega t}$. Since the kernel $\mathcal{K}$ and the probability measure $p(\omega)$ are real, we extract the real part of (2) and obtain:

$$\mathcal{K}(t_1, t_2) = E_\omega\big[\cos(\omega(t_1 - t_2))\big] = E_\omega\big[\cos(\omega t_1)\cos(\omega t_2) + \sin(\omega t_1)\sin(\omega t_2)\big]. \tag{3}$$

With this alternate expression of kernel function $\mathcal{K}$, the expectation term can be approximated by Monte Carlo integral [17]. Suppose we have $d$ samples $\omega_1, \dots, \omega_d$ drawn from $p(\omega)$, an estimate of our kernel $\mathcal{K}(t_1, t_2)$ can be constructed by $\frac{1}{d}\sum_{i=1}^{d} \cos(\omega_i t_1)\cos(\omega_i t_2) + \sin(\omega_i t_1)\sin(\omega_i t_2)$. As a consequence, Bochner's Theorem motivates the finite dimensional feature map to $\mathbb{R}^d$ via:

$$t \mapsto \Phi_d^{\mathcal{B}}(t) := \sqrt{\frac{1}{d}}\big[\cos(\omega_1 t), \sin(\omega_1 t), \dots, \cos(\omega_d t), \sin(\omega_d t)\big],$$

such that $\mathcal{K}(t_1, t_2) \approx \lim_{d \to \infty} \langle \Phi_d^{\mathcal{B}}(t_1), \Phi_d^{\mathcal{B}}(t_2) \rangle$.

So far we have obtained a specific functional form for $\Phi$, which is essentially a random projection onto the high-dimensional vector space of i.i.d random variables with density given by $p(\omega)$, where each coordinate is then transformed by trigonometric functions. However, it is not clear how to

| Feature maps specified by $\left[\phi_{2i}(t), \phi_{2i+1}(t)\right]$ | Origin | Parameters | Interpretations of $\omega$ |
|---|---|---|---|
| $\left[\cos\left(\omega_i(\mu)t\right), \sin\left(\omega_i(\mu)t\right)\right]$ | Bochner's | $\mu$: location-scale parameters specified for the *reparametrization trick*. | $\omega_i(\mu)$: converts the $i^{th}$ sample (drawn from auxiliary distribution) to target distribution under location-scale parameter $\mu$. |
| $\left[\cos\left(g_\theta(\omega_i)t\right), \sin\left(g_\theta(\omega_i)t\right)\right]$ | Bochner's | $\theta$: parameters for the inverse CDF $F^{-1} = g_\theta$. | $\omega_i$: the $i^{th}$ sample drawn from the auxiliary distribution. |
| $\left[\cos(\tilde{\omega}_i t), \sin(\tilde{\omega}_i t)\right]$ | Bochner's | $\{\tilde{\omega}\}_{i=1}^d$: transformed samples under non-parametric inverse CDF transformation. | $\tilde{\omega}_i$: the $i^{th}$ sample of the underlying distribution $p(\omega)$ in Bochner's Theorem. |
| $\left[\sqrt{c_{2i,k}}\cos(\omega_j t),\ \sqrt{c_{2i+1,k}}\sin(\omega_j t)\right]$ | Mercer's | $\{c_{i,k}\}_{i=1}^{2d}$: the Fourier coefficients of corresponding $\mathcal{K}_{\omega_j}$, for $j = 1, \ldots, k$. | $\omega_j$: the frequency for kernel function $\mathcal{K}_{\omega_j}$ (can be parameters). |

Table 1: The proposed functional forms of the feature map $\Phi = [\ldots, \phi_{2i}(t), \phi_{2i+1}(t), \ldots]$ motivated from Bochner's and Mercer's Theorem, with explanations of free parameters and interpretation of $\omega$.

sample from the unknown distribution of $\omega$. Otherwise we would already have $\mathcal{K}$ according to the Fourier transformation in (2). Mercer's Theorem, on the other hand, motivates a deterministic approach.

**Theorem 2** (Mercer's Theorem). *Consider the function class $L^2(\mathcal{X}, \mathbb{P})$ where $\mathcal{X}$ is compact. Suppose that the kernel function $\mathcal{K}$ is continuous with positive semidefinite and satisfy the condition $\int_{\mathcal{X}\times\mathcal{X}} \mathcal{K}^2(x, z)d\mathbb{P}(x)d\mathbb{P}(y) \leq \infty$, then there exist a sequence of eigenfunctions $(\phi_i)_{i=1}^\infty$ that form an orthonormal basis of $L^2(\mathcal{X}, \mathbb{P})$, and an associated set of non-negative eigenvalues $(c_i)_{i=1}^\infty$ such that*

$$\mathcal{K}(x, z) = \sum_{i=1}^\infty c_i \phi_i(x)\phi_i(z), \tag{4}$$

*where the convergence of the infinite series holds absolutely and uniformly.*

Mercer's Theorem provides intuition on how to embed instances from our functional domain $T$ into the infinite sequence space $\ell^2(\mathbb{N})$. To be specific, the mapping can be defined via $t \mapsto \Phi^{\mathcal{M}}(t) := \left[\sqrt{c_1}\phi_1(t), \sqrt{c_2}\phi_2(t), \ldots\right]$, and Mercer's Theorem guarantees the convergence of $\langle \Phi^{\mathcal{M}}(t_1), \Phi^{\mathcal{M}}(t_2) \rangle \to \mathcal{K}(t_1, t_2)$.

The two theorems have provided critical insight behind the functional forms of feature map $\Phi$. However, they are still not applicable. For the feature map motivated by Bochner's Theorem, let alone the infeasibility of sampling from unknown $p(\omega)$, the use of Monte Carlo estimation brings other uncertainties, i,e how many samples are needed for a decent approximation. As for the feature map from Mercer's Theorem, first of all, it is infinite dimensional. Secondly, it does not possess specific functional forms without making additional assumptions. The solutions to the above challenges are discussed in the next two sections.

## 4 Bochner Time Embedding

A practical solution to effectively learn the feature map suggested by Bochner's Theorem is to use the '*reparameterization trick*' [11]. Reparameterization trick provides ideas on sampling from distributions by using auxiliary variable $\epsilon$ which has known independent marginal distribution $p(\epsilon)$.

For 'location-scale' family distribution such as Gaussian distribution, suppose $\omega \sim N(\mu, \sigma)$, then with the auxiliary random variable $\epsilon \sim N(0, 1)$, $\omega$ can be reparametrized as $\mu + \sigma\epsilon$. Now samples of $\omega$ are transformed from samples of $\epsilon$, and the free distribution parameters $\mu$ and $\sigma$ can be optimized

as part of the whole learning model. With Gaussian distribution, the feature map $\Phi_d^{\mathcal{B}}$ suggested by Bochner's Theorem can be effectively parameterized by $\mu$ and $\sigma$, which are also the inputs to the functions $\omega_i(\mu, \sigma)$ that transforms the $i^{th}$ sample from the auxiliary distribution to a sample of target distribution (Table 1). A potential concern here is that the 'location-scale' family may not be rich enough to capture the complexity of temporal patterns under Fourier transformation. Indeed, the Fourier transform of a Gaussian function in the form of $f(x) \equiv e^{-ax^2}$ is another Gaussian function. An alternate approach is to use inverse cumulative distribution function CDF transformation.

Let $F^{-1}$ be the inverse CDF of some probability distribution (if exists), then for $\epsilon$ sampled from uniform distribution, we can always use $F^{-1}(\epsilon)$ to generate samples of the desired distribution. This suggests parameterizing the inverse CDF function as $F^{-1} \equiv g_\theta(.)$ with some functional approximators such as neural networks or *flow-based* CDF estimation methods including *normalizing flow* [18] and *RealNVP* [5] (see the Appendix for more discussions). As a matter of fact, if the samples are first drawn (following either transformation method) and held fixed during training, we can consider using non-parametric transformations. For $\{\omega_i\}_{i=1}^d$ sampled from auxiliary distribution, let $\tilde{\omega}_i = F^{-1}(\omega_i), i = 1, 2, \cdot, d$, for some non-parametric inverse CDF $F^{-1}$. Since $\omega_i$ are fixed, learning $F^{-1}$ amounts to directly optimize the transformed samples $\{\tilde{\omega}\}_{i=1}^d$ as free parameters.

In short, the Bochner's time feature maps can be realized with *reparametrization trick* or parametric/nonparametric inverse CDF transformation. We refer to them as Bochner time encoding. In Table 1, we conclude the functional forms for Bochner time encoding and provides explanations of the free parameters as well as the meanings of $\omega$. A sketched visual illustration is provided in the left panel of Table 2. Finally, we provide the theoretical justification that with samples drawn from the corresponding distribution $p(w)$, the Monte Carlo approximation converges uniformly to the kernel function $\mathcal{K}$ with high probability. The upper bound stated in Claim 1 provides some guidelines for the number of samples needed to achieve a good approximation.

**Claim 1.** *Let $p(\omega)$ be the corresponding probability measure stated in Bochner's Theorem for kernel function $\mathcal{K}$. Suppose the feature map $\Phi$ is constructed as described above using samples $\{\omega_i\}_{i=1}^d$, we have*

$$Pr\Big( \sup_{t_1, t_2 \in T} \big| \Phi_d^{\mathcal{B}}(t_1)^{'} \Phi_d^{\mathcal{B}}(t_2) - \mathcal{K}(t_1, t_2) \big| \geq \epsilon \Big) \leq 4\sigma_p \sqrt{\frac{t_{\max}}{\epsilon}} exp\Big( \frac{-d\epsilon^2}{32} \Big), \qquad (5)$$

*where $\sigma_p^2$ is the second momentum with respect to $p(\omega)$.*

The proof is provided in supplement material.

Therefore, we can use $\Omega\big( \frac{1}{\epsilon^2} \log \frac{\sigma_p^2 t_{\max}}{\epsilon} \big)$ samples (at the order of hundreds if $\epsilon \approx 0.1$) from $p(\omega)$ to have $\sup_{t_1, t_2 \in T} \big| \Phi_d^{\mathcal{B}}(t_1)^{'} \Phi_d^{\mathcal{B}}(t_2) - \mathcal{K}(t_1, t_2) \big| < \epsilon$ with any probability.

## 5 Mercer Time Embedding

Mercer's Theorem solves the challenge of embedding time span onto a sequence space, however, the functional form of $\Phi$ is unknown and the space is infinite-dimensional. To deal with the first problem, we need to make an assumption on the periodic properties of $\mathcal{K}$ to meet the condition in Proposition 1, which states a fairly straightforward formulation of the functional mapping $\Phi(.)$.

**Proposition 1.** *For kernel function $\mathcal{K}$ that is continuous, PSD and translation-invariant with $\mathcal{K} = \psi(t_1 - t_2)$, suppose $\psi$ is a even periodic function with frequency $\omega$, i.e $\psi(t) = \psi(-t)$ and $\psi\big(t + \frac{2k}{\omega}\big) = \psi(t)$ for all $t \in [-\frac{1}{\omega}, \frac{1}{\omega}]$ and integers $k \in \mathbb{Z}$, the eigenfunctions of $\mathcal{K}$ are given by the Fourier basis.*

The proof of Proposition 1 is provided in supplement material.

Notice that in our setting the kernel $\mathcal{K}$ is not necessarily periodic. Nonetheless we may assume that the temporal patterns can be detected from a finite set of periodic kernels $\mathcal{K}_\omega : T \times T \to \mathbb{R}, \omega \in \{\omega_1, \ldots, \omega_k\}$, where each $\mathcal{K}_\omega$ is a continuous, translation-invariant and PSD kernel further endowed with some frequency $\omega$. In other words, we project the unknown kernel function $\mathcal{K}$ onto a set of periodic kernels who have the same properties as $\mathcal{K}$.

According to Proposition 1 we immediately see that for each periodic kernel $\mathcal{K}_{\omega_i}$ the eigenfunctions stated in Mercer's Theorem are given by: $\phi_{2j}(t) = 1$, $\phi_{2j}(t) = \cos\big(\frac{j\pi t}{\omega_i}\big), \phi_{2j+1}(t) =$

$\sin\left(\frac{j\pi t}{\omega_i}\right)$ for $j = 1, 2, \ldots$, with $c_i, i = 1, 2, \ldots$ giving the corresponding Fourier coefficients. Therefore we have the infinite dimensional Mercer's feature map for each $\mathcal{K}_\omega$:

$$t \mapsto \Phi_\omega^{\mathcal{M}}(t) = \left[\sqrt{c_1}, \ldots, \sqrt{c_{2j}}\cos\left(\frac{j\pi t}{\omega}\right), \sqrt{c_{2j+1}}\sin\left(\frac{j\pi t}{\omega}\right), \ldots\right],$$

where we omit the dependency of all $c_j$ on $\omega$ for notation simplicity.

One significant advantage of expressing $K_\omega$ by Fourier series is that they often have nice truncation properties, which allows us to use the truncated feature map without loosing too much information. It has been shown that under mild conditions the Fourier coefficients $c_j$ decays exponentially to zero [21], and classic approximation theory guarantees a uniform convergence bound for truncated Fourier series [9] (see Appendix for discussions). As a consequence, we propose to use the truncated feature map $\Phi_{\omega,d}^{\mathcal{M}}(t)$, and thus the complete Mercer's time embedding is given by:

$$t \mapsto \Phi_d^{\mathcal{M}} = \left[\Phi_{\omega_1,d}^{\mathcal{M}}(t), \ldots, \Phi_{\omega_k,d}^{\mathcal{M}}(t)\right]^\top. \tag{6}$$

Therefore Mercer's feature map embeds the periodic kernel function into the high-dimensional space spanned by truncated Fourier basis under certain frequency. As for the unknown Fourier coefficients $c_j$, it is obvious that learning the kernel functions $\mathcal{K}_\omega$ is in form equivalent to learning their corresponding coefficients. To avoid unnecessary complications, we treat $c_j$ as free parameters.

Last but not least, we point out that the set of frequencies $\{\omega_1, \ldots, \omega_k\}$ that specifies each periodic kernel function should be able to cover a broad range of bandwidths in order to capture various signals and achieve good approximation. They can be either fixed or jointly optimized as free parameters. In our experiments they lead to similar performances if properly initialized, such as using a geometrically sequence: $\omega_i = \omega_{\max} - (\omega_{\max} - \omega_{\min})^{i/k}, i = 1, \ldots, k$, to cover $[\omega_{\min}, \omega_{\max}]$ with a focus on high-frequency regions. The sketched visual illustration is provided in the right panel of Table 2.

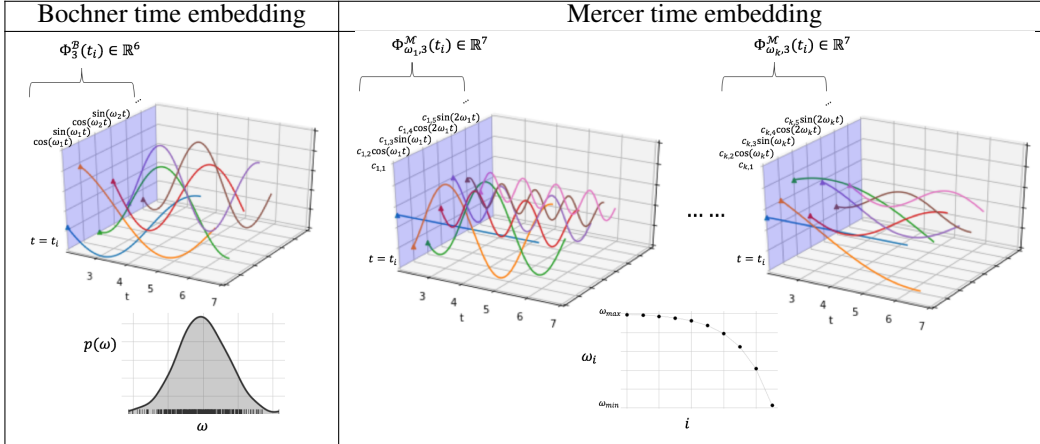

Table 2: Sketched visual illustration of the proposed Bochner and Mercer time embedding ($\Phi_d^{\mathcal{B}}(t)$ and $\Phi_{\omega,d}^{\mathcal{M}}(t)$) for a specific $t = t_i$ with $d = 3$. In right panel the scale of sine and cosine waves decreases as their frequency gets larger, which is a common phenomenon for Fourier series.

## 6 Time-event Interaction

Learning time-event interaction is crucial for continuous-time event sequence prediction. After embedding time span into finite-dimensional vector spaces, we are able to directly model interactions using time and event embeddings. It is necessary to first project the time and event representations onto the same space. For an event sequence $\{(e_1, t_1), \ldots, (e_q, t_q)\}$ we concatenate the event and time representations into $[\mathbf{Z}, \mathbf{Z}_T]$ where $\mathbf{Z} = [Z_1, \ldots, Z_q]$, $\mathbf{Z}_T = [\Phi(t_1), \ldots, \Phi(t_q)]$ and project them into the query, key and value spaces. For instance, to consider only linear combinations of event and time representations in query space, we can simply use $\mathbf{Q} = [\mathbf{Z}, \mathbf{Z}_T]\mathbf{W}_0 + b_0$. To capture non-linear relations hierarchically, we may consider using multilayer perceptrons (MLP) with activation functions, such as

$$\mathbf{Q} = \text{ReLU}\big([\mathbf{Z}, \mathbf{Z}_T]\mathbf{W}_0 + b_0\big)\mathbf{W}_1 + b_1,$$

where ReLU(.) is the rectified linear unit. Residual blocks can also be added to propagate useful lower-level information to final output. When predicting the next time-dependent event $(e_{q+1}, t_{q+1})$, to take account of the time lag between each event in input sequence and target event we let $\tilde{t}_i = t_{q+1} - t_i, i = 1, \ldots, q$ and use $\Phi(\tilde{t}_i)$ as time representations. This does not change the relative time difference between input events, i.e. $\tilde{t}_i - \tilde{t}_j = t_i - t_j$ for $i, j = 1, \ldots, q$, and now the attention weights and prediction becomes a function of next occurrence time.

# 7 Experiment and Result

We evaluate the performance of the proposed time embedding methods with self-attention on several real-world datasets from various domains. The experiemnts aim at quantitatively evaluating the performance of the four time embedding methods, and comparing them with baseline models.

## 7.1 Data Sets

- **Stack Overflow**[2] dataset records user's history awarded badges in a question-answering website. The task is to predict the next badge the user receives, as a classification task.
- **MovieLens**[3] is a public dataset consists of movie rating for benchmarking recommendations algorithms [7]. The task is to predict the next movie that the user rates for recommendation.
- **Walmart.com dataset** is obtained from Walmart's online e-commerce platform in the U.S[4]. It contains the session-based search, view, add-to-cart and transaction information with timestamps for each action from selected users. The task is to predict the next-view item for recommendation. Details for all datasets are provided in supplemnetary meterial.

**Data preparation** - For fair comparisons with the baselines, on the *MovieLens* dataset we follow the same prepossessing steps mentioned in [10]. For users who rated at least three movies, we use their second last rating for validation and their last rated movie for testing. On the *stack overflow* dataset we use the same filtering procedures described in [12] and randomly split the dataset on users into training (80%), validation (10%) and test (10%). On the *Walmart.com* dataset we filter out users with less than ten activities and products that interacted with less than five users. The training, validation and test data are splited based on session starting time chronically.

## 7.2 Baselines and Model configurations

We compare the proposed approach with LSTM, the time-aware RNN model (*TimeJoint*) [12] and recurrent marked temporal point process model (*RMTPP*) [6] on the *Stack Overflow* dataset. We point out that the two later approaches also utilize time information. For the above three models, we reuse the optimal model configurations and metrics (classification *accuracy*) reported in [12] for the same *Stack Overflow* dataset.

For the recommendation tasks on *MovieLens* dataset, we choose the seminal session-based RNN recommendation model (*GRU4Rec*) [8], convolutional sequence embedding method (*Caser*) [19] and translation-based recommendation model (*TransRec*) [10] as baselines. These position-aware sequential models have been shown to achieve cutting-edge performances on the same *MovieLens* dataset [10]. We also reuse the metrics - top K hitting rate (*Hit@K*) and normalized discounted cumulative gain (*NDCG@K*), as well as the optimal model configurations reported in [10].

On the *Walmart.com* dataset, other than *GRU4Rec* and *TransRec*, we compare with an attention-based RNN model *RNN+attn*. The hyper-parameters of the baselines are tuned for optimal performances according to the *Hit@10* metric on the validation dataset. The outcomes are provided in Table 3.

As for the proposed time embedding methods, we experimented on the Bochner time embedding with the *reparameterization trick* using normal distribution (*Bochner Normal*), the parametric inverse CDF transformation (*Bochner Inv CDF*) with MLP, MLP + residual block, masked autoregressive flow (*MAF*) [16] and non-volume preserving transformations (*NVP*) [5], the non-parametric inverse CDF transformation (*Bochner Non-para*), as well as the Mercer time embedding. For the purpose

of *ablation study*, we compare with the original positional encoding self-attention (*PosEnc*) for all tasks (Table 3). We use $d = 100$ for both Bochner and Mercer time embedding, with the sensitivity analysis on time embedding dimensions provided in appendix. We treat the dimension of Fourier basis $k$ for Mercer time embedding as hyper-parameter, and select from $\{1, 5, 10, 15, 20, 25, 30\}$ according to the validation *Hit@10* as well. When reporting the results in Table 3, we mark the model configuration that leads to the optimal validation performance for each of our time embedding methods. Other configurations and training details are provided in appendix.

## 7.3   Experimental results

| | | | | Stack Overflow | | | | |
|---|---|---|---|---|---|---|---|---|
| **Method** | LSTM | TimeJoint | RMTPP | PosEnc | **Bochner Normal** | **Bochner Inv CDF** | **Bochner Non-para** | **Mercer** |
| **Accuracy** | 46.03(.21) | 46.30(.23) | 46.23(.24) | 44.03(.33) | 44.89(.46) | 44.67(.38) | 46.27(0.29) | **46.83(0.20)** |
| config | | | | | | *NVP* | | $k = 10$ |
| | | | | MovieLens-1m | | | | |
| **Method** | GRU4Rec | Caser | TransRec | - | - | - | - | - |
| **Hit@10** | 75.01(.25) | 78.86(.22) | 64.15(.27) | 82.45(.31) | 81.60(.69) | 82.52(.36) | 82.86(.22) | **82.92 (.17)** |
| **NDCG@10** | 55.13(.14) | 55.38(.15) | 39.72(.16) | 59.05(.14) | 59.47(.56) | 60.80(.47) | 60.83(.15) | **61.67 (.11)** |
| config | | | | | | *MAF* | | $k = 5$ |
| | | | | Walmart.com data | | | | |
| **Method** | GRU4Rec | RNN+attn | TransRec | - | - | - | - | - |
| **Hit@5** | 4.12(.19) | 5.90(.17) | 7.03(.15) | 8.63(.16) | 4.27(.91) | 9.04(.31) | 9.25(.15) | **10.92(.13)** |
| **NDCG@5** | 4.03(.20) | 4.66(.17) | 5.62(.17) | 6.92(.14) | 4.06(.94) | 7.27(.26) | 7.34(.12) | **8.90(.11)** |
| **Hit@10** | 6.71(.50) | 9.03(.44) | 10.38(.41) | 12.49(.38) | 7.66(.92) | 12.77(.65) | 13.16(.41) | **14.94(.31)** |
| **NDCG@10** | 4.97(.31) | 7.36(.26) | 8.72(.26) | 10.84(.26) | 6.02(.99) | 10.95(.74) | 11.36(.27) | **12.81(.22)** |
| config | | | | | | *MAF* | | $k = 25$ |

Table 3: Performance metrics for the proposed apporach and baseline models. All results are converted to percentage by multiplying by 100, and the standard deviations computed over ten runs are given in the parenthesis. The proposed methods and the best outcomes are highlighted in bold font. The *config* rows give the optimal model configuration for *Bochner Inv CDF* (among using MLP, MLP + redisual block, *MAF* and *NVP* as CDF learning method) and *Mercer* (among $k = 1, 5, \ldots, 30$).

We observe in Table 3 that the proposed time embedding with self-attention compares favorably to baseline models on all three datasets. For the *Stack Overflow* and *Walmart.com* dataset, *Mercer* time embedding achieves best performances, and on *MovieLens* dataset the *Bochner Non-para* outperforms the remaining methods. The results suggest the effectiveness of the functional time representation, and the comparison with positional encoding suggests that time embedding are more suitable for continuous-time event sequence modelling. On the other hand, it appears that *Bochner Normal* and *Bochner Inv CDF* has higher variances, which might be caused by their need for sampling steps during the training process. Otherwise, *Bochner Inv CDF* has comparable performances to *Bochner Non-para* across all three datasets. In general, we observe better performances from *Bochner Non-para* time embedding and Mercer time embedding. Specifically, with the tuned Fourier basis degree $k$, Mercer's method consistently outperforms others across all tasks. While $d$, the dimension of time embedding, controls how well the bandwidth of $[\omega_{\min}, \omega_{\max}]$ is covered, $k$ controls the degree of freedom for the Fourier basis under each frequency. When $d$ is fixed, larger $k$ may lead to overfitting issue for the time kernels under certain frequencies, which is confirmed by the sensitivity analysis on $k$ provided in Figure 1b.

In Figure 2, we visualize the average attention weights across the whole population as functions of time and user action or product department on the Walmart.com dataset, to demonstrate some of the useful temporal patterns captured by the Mercer time embedding. For instance, Figure 2a shows that when recommending the next product, the model learns to put higher attention weights on the last searched products over time. Similarly, the patterns in Figure 2b indicate that the model captures the signal that customers often have prolonged or recurrent attentions on *baby products* when compared with *electronics* and *accessories*. Interestingly, when predicting the attention weights by using future time points as input (Figure 2c), we see our model predicts that the users almost completely lose attention on their most recent purchased products (which is reasonable), and after a more extended period none of the previously interacted products matters anymore.

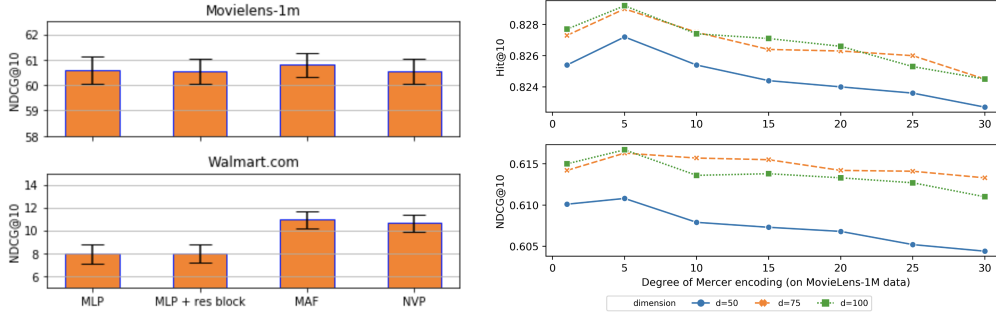

Figure 1: (a). We show the results of *Bochner Inv CDF* on the *Movielens* and *Walmart.com* dataset with different distributional learning methods. (b). The sensitivity analysis on Mercer time encoding on the *Movielens* dataset by varying the degree of Fourier basis $k$ under different dimension $d$.

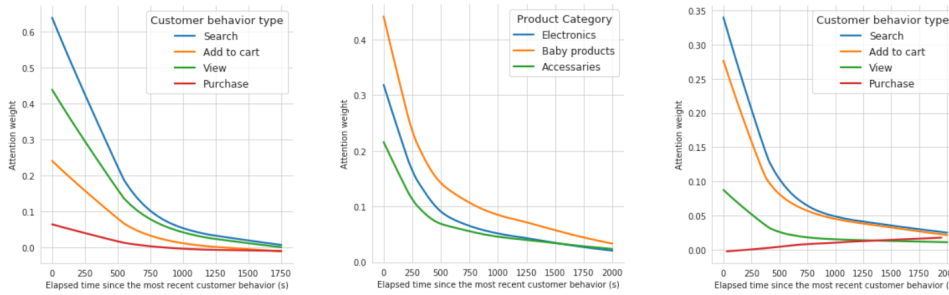

(a) The temporal patterns in aver-age attention weight decay on the last interacted product after differ-ent user actions, as time elapsed.

(b) The temporal patterns in aver-age attention weight decay on the last viewed product from different departments, as time elapsed.

(c) The prediction of future atten-tion weight on the last interacted product as a function of time and different user actions.

Figure 2: Temporal patterns and time-event interactions captured by time and event representations on the Walmart.com dataset.

**Discussion**. By employing state-of-the-art CDF learning methods, *Bochner Inv CDF* achieves better performances than positional encoding and other baselines on Movlielens and Walmart.com dataset (Figure 1a). This suggests the importance of having higher model complexity for learning the $p(\omega)$ in Bochner's Thm, and also explains why *Bochner Normal* fails since normal distribution has limited capacity in capturing complicated distributional signals. On the other hand, *Bochner Non-para* is actually the special case of *Mercer*'s method with $k = 1$ and no intercept. While Bochner's methods originate from random feature sampling, Mercer's method grounds in functional basis expansion. In practice, we may expect Mercer's method to give more stable performances since it does not rely on distributional learning and sampling. However, with advancements in Bayesian deep learning and probabilistic computation, we may also expect *Bochner Inv CDF* to work appropriately with suitable distribution learning models, which we leave to future work.

## 8  Conlusion

We propose a set of time embedding methods for functional time representation learning, and demonstrate their effectiveness when using with self-attention in continuous-time event sequence prediction. The proposed methods come with sound theoretical justifications, and not only do they reveal temporal patterns, but they also capture time-event interactions. The proposed time embedding methods are thoroughly examined by experiments using real-world datasets, and we find Mercer time embedding and Bochner time embedding with non-parametric inverse CDF transformation giving superior performances. We point out that the proposed methods extend to general time representation learning, and we will explore adapting our proposed techniques to other settings such as temporal graph representation learning and reinforcement learning in the our future work.

## Footnotes

[2]https://archive.org/details/stackexchange

[3]https://grouplens.org/datasets/movielens/1m/

[4]https://www.walmart.com

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
