[Supplementary Material 1 · appendix.pdf]

# A  Appendix

## A.1  Proof of Claim 1

*Proof.* Define the score $S(t_1, t_2) = \Phi_d^{\mathcal{B}}(t_1)' \Phi_d^{\mathcal{B}}(t_2)$. The goal is to derive a uniform upper bound for $s(t_1, t_2) - \mathcal{K}(t_1, t_2)$. By assumption $S(t_1, t_2)$ is an unbiased estimator for $\mathcal{K}(t_1, t_2)$, i.e. $E[S(t_1, t_2)] = \mathcal{K}(t_1, t_2)$. Due to the translation-invariant property of $S$ and $\mathcal{K}$, we let $\Delta(t) \equiv s(t_1, t_2) - \mathcal{K}(t_1, t_2)$, where $t \equiv t_1 - t_2$ for all $t_1, t_2 \in [0, t_{\max}]$. Also we define $s(t_1 - t_2) := S(t_1, t_2)$. Therefore $t \in [-t_{\max}, t_{\max}]$, and we use $t \in \tilde{T}$ as the shorthand notation. The LHS in (1) now becomes $\Pr\big(\sup_{t \in \tilde{T}} |\Delta(t)| \geq \epsilon\big)$.

Note that $\tilde{T} \subseteq \cup_{i=0}^{N-1} T_i$ with $T_i = \big[ - t_{\max} + \frac{2it_{\max}}{N}, -t_{\max} + \frac{2(i+1)t_{\max}}{N} \big]$ for $i = 1, \ldots, N$. So $\cup_{i=0}^{N-1} T_i$ is a finite cover of $\tilde{T}$. Define $t_i = -t_{\max} + \frac{(2i+1)t_{\max}}{N}$, then for any $t \in T_i$, $i = 1, \ldots, N$ we have

$$
\begin{aligned}
\big|\Delta(t)\big| &= \big|\Delta(t) - \Delta(t_i) + \Delta(t_i)\big| \\
&\leq \big|\Delta(t) - \Delta(t_i)\big| + \big|\Delta(t_i)\big| \\
&\leq L_\Delta \big|t - t_i\big| + \big|\Delta(t_i)\big| \\
&\leq L_\Delta \frac{2t_{\max}}{N} + \big|\Delta(t_i)\big|,
\end{aligned}
\tag{7}
$$

where $L_\Delta = \max_{t \in \tilde{T}} \big\|\nabla\Delta(t)\big\|$ (since $\Delta$ is differentiable) with the maximum achieved at $t^*$. So we may bound the two events separately.

For $|\Delta(t_i)|$ we simply notice that trigeometric functions are bounded between $[-1, 1]$, and therefore $-1 \leq \Phi_d^{\mathcal{B}}(t_1)' \Phi_d^{\mathcal{B}}(t_2) \leq 1$. The Hoeffding's inequality for bounded random variables immediately gives us:

$$
\Pr\big(|\Delta(t_i)| > \frac{\epsilon}{2}\big) \leq 2 \exp\Big( -\frac{d\epsilon^2}{16} \Big).
$$

So applying the Hoeffding-type union bound to the finite cover gives

$$
\Pr\big( \cup_{i=0}^{N-1} |\Delta(t_i)| \geq \frac{\epsilon}{2} \big) \leq 2N \exp\Big( -\frac{d\epsilon^2}{16} \Big)
\tag{8}
$$

For the other event we first apply Markov inequality and obtain:

$$
\Pr\big(L_\Delta \frac{2t_{\max}}{N} \geq \frac{\epsilon}{2}\big) = \Pr\big(L_\Delta \geq \frac{\epsilon N}{4t_{\max}}\big) \leq \frac{4t_{\max} E[L_\Delta^2]}{\epsilon N}.
\tag{9}
$$

Also, since $E\big[s(t_1 - t_2)\big] = \psi(t_1 - t_2)$, we have

$$
E\big[L_\Delta^2\big] = E\big\|\nabla s(t^*) - \nabla\psi(t^*)\big\|^2 = E\big\|\nabla s(t^*)\big\|^2 - E\big\|\nabla\psi(t^*)\big\|^2 \leq E\big\|\nabla s(t^*)\big\|^2 = \sigma_p^2,
\tag{10}
$$

where $\sigma_p^2$ is the second momentum with respect to $p(\omega)$.

Combining (8), (9) and (8) gives us:

$$
\Pr\Big( \sup_{t \in \tilde{T}} |\Delta(t)| \geq \epsilon \Big) \leq 2N \exp\Big( -\frac{d\epsilon^2}{16} \Big) + \frac{4t_{\max}\sigma_p^2}{\epsilon N}.
\tag{11}
$$

It is straightforward to examine that the RHS of (11) is a convex function of $N$ and is minimized by $N^* = \sigma_p \sqrt{\frac{2t_{\max}}{\epsilon}} \exp\big(\frac{d\epsilon^2}{32}\big)$. Plug $N^*$ back to (11) and we obtain bound stated in Claim 1. $\qquad\square$

## A.2  Proof of Proposition 1

*Proof.* We first define the kernel linear operator $\mathcal{T}_\mathcal{K}$ on $L^2(T)$ via $\mathcal{T}_\mathcal{K}(f)(t_1) = \int_T f(t_1)\mathcal{K}(t_1, t_2)d\mathbb{P}(t_2)$, where $\mathbb{P}$ is a non-negative measure over $T = [0, t_{\max}]$. For notation simplicity we do not explicitly index $\mathcal{K}$ with its frequency. The more complete statement of Mercer's Theorem is that under the conditions specified in Theorem 2,

$$
\mathcal{T}_\mathcal{K}(\phi_j) = c_j \phi_j \quad \text{for } j = 1, 2, \ldots,
\tag{12}
$$

which leads to the representation of the kernel as $\mathcal{K}(x, z) = \sum_{i=1}^{\infty} c_i \phi_i(x) \phi_i(z)$.

Therefore we only need to show the Fourier basis gives the eigenfunctions of the kernel linear operator $\mathcal{T}_{\mathcal{K}}$. Without loss of generality we assume the frequency is $2\pi$, i.e. $\psi$ is a even periodic function on $[-1, 1]$ and extend to the real line by $\psi(t + 2k) = \psi(t)$ for $t \in [-1, 1]$. So now the kernel linear operator is expressed by:

$$\mathcal{T}_{\mathcal{K}}(f)(t_1) = \int_{-1}^{1} \psi(t_1 - t_2) f(t_2) dt_2. \tag{13}$$

Now we show that the eigenfunctions for the kernel linear operator are given by Fourier basis. Suppose $\phi_{2j}(t) = \cos(\pi j t)$ for $j = 1, 2, \ldots$, we have

$$
\begin{aligned}
\mathcal{T}_{\mathcal{K}}(\phi_{2j})(t_1) &= \int_{-1}^{1} \psi(t_1 - t_2) \cos(2\pi t_2) dt_2 \\
&\overset{(a)}{=} \int_{-1}^{1} \psi(t_3) \cos\big(2\pi j(t_1 + t_3)\big) dt_3 \quad \text{(with } t_3 = t_1 - t_2) \\
&\overset{(b)}{=} \cos(\pi j t_1) \int_{-1}^{1} \psi(t_3) \cos(\pi j t_3) dt_3 - \sin(\pi j t_1) \int_{-1}^{1} \psi(t_3) \sin(\pi j t_3) dt_3 \\
&\overset{(c)}{=} c_j \cos(\pi j t_1),
\end{aligned}
\tag{14}
$$

where in $(a)$ we use a change of variable and utilize the periodic property of $\psi$ and the cosine function. In $(b)$ we apply the sum formula of trigonometric functions, and in $(c)$ we simply use the fact that $\int_{-1}^{1} \phi(t_3) \sin(\pi j t_3) dt_3 = 0$ because $\psi$ is an even function. Similar arguments show that $\phi_{2j+1}(t) = \sin(\pi j t)$ for $j = 1, 2, \ldots$ are also the eigenfunctions for $\mathcal{T}_{\mathcal{K}}$. Since the Fourier basis form a complete orthonormal basis of $L^2(T)$, according to the complete Mercer's Theorem we see that the eigenfunctions of $\mathcal{K}$ are exactly given by the Fourier basis. □

### A.3  Fourier series under truncation

In this part we briefly discuss the exponential decay for the eigenvalues $c_j$ and the uniform bound on approximation error for truncated Fourier series mentioned in Section 5. Notice that Bochner's Theorem also applies to the periodic kernels ($\mathcal{K}(t_1, t_2) \equiv \psi(t_1 - t_2)$) stated in Mercer time embedding, such that

$$\psi(t_1 - t_2) = \lambda \int e^{-i(t_1 - t_2)\omega} p(\omega),$$

where $\lambda$ is the scaling constant such that $p(\omega)$ is a probability measure. It has been shown that if $\log p(\omega) \asymp -\omega^a - \log(\lambda)$ for some $a > 1$, then there is a constant $b$ such that the Fourier coefficients satisfy: $c_j \asymp e^{-bj \log j}$ as $j \to \infty$ [21].

As for the approximation error of truncated Fourier series, first we use $S_d(t_1 - t_2)$ to denote the partial sum of the Fourier series for $\psi(t_1 - t_2)$ up to the $d^{th}$ order. According to the Corollary I in [9], if $\psi$ is $\ell-Lipschitz$, then we have the uniform convergence bound

$$\big|\psi(t_1 - t_2) - S_d(t_1 - t_2)\big| \leq \frac{C\ell \log d}{d}$$

for all $t_1, t_2 \in T$ under some constant $C$.

The above classical results suggest the exponential decay of the Fourier coefficients as well as the uniform convergence property of the truncated Fourier series and further validate the Mercer time embedding.

### A.4  Flow-based distribution learning

Here we briefly introduce the idea of constructing and sampling from an arbitrarily complex distribution from a known auxiliary distribution by a sequence of invertible transformations. It is motivated by the basic change of variable theorem, which we state below.

Figure 3: Illustration of the distribution transformation flow.

Given an auxiliary random variable $\mathbf{z}$ following some know distribution $q(z)$, suppose another random variable $\mathbf{x}$ is constructed via a one-to-one mapping from $\mathbf{z}$: $\mathbf{x} = f(\mathbf{z})$, then the density function of $\mathbf{x}$ is given by:

$$p(x) = q(z)\left|\frac{dz}{dx}\right| = q\big(f^{-1}(x)\big)\left|\frac{df^{-1}}{dx}\right|. \tag{15}$$

We can parameterize the one-to-one function $f(.)$ with free parameters $\theta$ and optimize them over the observed evidence such as by maximizing the log-likelihood. By stacking a sequence of $Q$ one-to-one mappings, i.e. $\mathbf{x} = f_Q \circ f_{Q-1} \circ \ldots f_1(\mathbf{z})$, we can construct complicated density functions. It is easy to show by chaining that $p(x)$ is given by:

$$\log p(\mathbf{x}) = \log q(\mathbf{z}) - \sum_{i=1}^{Q}\left|\frac{df^{-1}}{dz_i}\right|. \tag{16}$$

A sketched graphical illustration of the concept is shown in Figure 3.

Similarity, samples from the auxiliary distribution can be transformed to the unknown target distribution in the same manner, and the transformed samples are essentially parameterized by the transformation mappings, i.e. the $g_\theta(\omega_i)$ in the second row of Table 1.

## A.5 Dataset details

The *Stach Overflow* dataset contains 6,000 users and 480,000 events of awarding badges. Timestamps are provided when a user is awarded a badge. There are 22 unique badges after filtering, and the prediction of the next badge is treated as a classification task. Event sequences are generated with the same procedures described in [12].

The *MovieLens* dataset, which is a benchmark for evaluating collaborative filtering algorithms, consists of 60,40 users and 3,416 movies with a total of one million ratings. The implicit feedback of rating actions characterizes user-movie interactions. Therefore the event sequence for each user is not a complete observation for their watching records. To construct event sequences from observations, we follow the same steps as described in [10], where for each user, the final rating is used for testing, the second to last rating is used for validations, and the remaining sequence is used as the input sequence.

In the *Walmart.com* dataset, there are about 72,000 users and about 1.7 million items with user-item interactions characterized by search, view, add-to-cart and transaction (purchase). The product catalog information is also available, which provides the name, brand and categories for each product. User activity records are aggregated in term of online shopping sessions. So for each user session, we construct event sequences using the same steps as the *MovieLens* dataset, in a sequence-to-sequence fashion.

## A.6 Training and model configuration

We select the number of blocks among {1,2,3} and the number of attention heads among {1,2,3,4} for each dataset according to their validation performances. We do not experiment on using dropout or regularizations unless otherwise specified. We use the default settings for *MAF* and *NVP* provided by TensorFlow Probability [5][6] when learning the distribution for *Bochner Inv CDF*. Notice that we have not carefully tuned the *MAF* and *NVP* for *Bochner Inv CDF*, since our major focus is to show the validity of these approaches.

**Stack Overflow** - For all the models that we implement, following the baseline settings reported in [12], the hidden dimension for event representations is set to 32. The dimensions of time embeddings are also set to be 32. In each self-attention block, we concatenate time embeddings to event embeddings and project them to key, query and value spaces through linear projections, i.e.

$$\mathbf{Q} = [\mathbf{Z}, \mathbf{Z}_T]\mathbf{W}_Q, \ \mathbf{K} = [\mathbf{Z}, \mathbf{Z}_T]\mathbf{W}_K, \ \mathbf{V} = [\mathbf{Z}, \mathbf{Z}_T]\mathbf{W}_V,$$

where $\mathbf{Z}$ and $\mathbf{Z}_T$ are the entity and time embeddings, $\mathbf{W}_Q$, $\mathbf{W}_K$, $\mathbf{W}_V$ are the projection matrices. We find that using a larger hidden dimension with to many attention blocks quickly leads to over-fit in this dataset, and using the single-head self-attention gives best performances. Therefore we end up using only one self-attention block. The maximum length of the event sequence is set to be 100. For the classification problem, we feed the output sequence embeddings into a fully connected layer to predict the logits for each class and use the softmax function to compute cross-entropy loss.

**MovieLens** - We adopt the self-attention model architecture used by the baseline models [10] for fair comparisons by replacing the positional encoding with our time embedding. To be specific, the dimension for event representation is set to 50, the number of attention blocks is two and only one head is used in each attention block. To be consistent with the positional encoding self-attention baseline reported in [10], we set the dropout rate to 0.2 and the l2 regularization to 0. The maximum length of the sequence is 200, and the batch size is 128. We also adopt the shared embeddings idea for event representations [10], where we use the same set of parameters for the event embeddings layers and the final softmax layers. Finally, the cross-entropy loss with negative sampling is used to speed up the training process.

**Walmart.com dataset** - Given the massive number of items in the dataset, we first train a shallow embeddings model to learn coarse item representations according to their context features and use those embeddings as initialization for the product representations in our model [24]. The dimension of item embedding and time embedding are set to 100. Each user action (search, view, add-to-cart, transaction) is treated as a token and has a 50-dimensional vector representation that is jointly optimized as part of the model. The action embedding is concatenated to the time-event representations and together they give the time-event-action embedding. To capture time-event and time-action interactions, we first project the joint embeddings onto query, key and value spaces also with linear projections as we did on the *Stack Overflow* dataset. We find that using two attention blocks and a single head give the best results. Since the task is to predict the next-view item in the same session, we also use the cross-entropy loss with negative sampling.

During training, we apply the *early stopping* where we terminate the model training if the validation performance has not increased for 10 epochs. When training models for predicting the next events, we refer to the *masked self-attention* training procedure proposed in [20] to prevent information leak while maintaining a fast training speed.

### A.6.1  Initialization for time embedding methods

For the *Bochner Normal* method, we use the standard normal distribution as initial distribution in all experiments. The parametric inverse CDF function for the *Bochner Inv CDF* is carried out by a three-layer MLP under uniform initialization. For the *Bochner Non-para* method, since each $\phi_i(t) = \sin(\omega_i t)$ or $\phi_i(t) = \cos(\omega_i t)$, they have a period of $2\pi/\omega_i$. Since we would like $\phi_i$ to capture underlying temporal signals, the scale of the potential periodicity in the experimented dataset should be taken into consideration. For instance, on the *Stack Overflow* dataset, it can take days or weeks before the next event happens. Therefore, if the temporal signals were to have underlying periods, it should be on the scale from several days to several weeks. For the *Walmart.com* dataset, the next activities are often operated within minutes. Therefore the periods should range from seconds to hours.

Therefore, in our experiments, we set the frequencies to cover a suitable range of period $[\tau_{\min}, \tau_{\max}]$ where $\tau \equiv 1/\omega$. With out loss of generality, the $\tau_{\min}$ and $\tau_{\max}$ are based on the minimum and maximum time span between consecutive events observed in data. We find that using geometric sequences that cover $[\tau_{\min}, \tau_{\max}]$ as initialization gives better results than random sampling and equal spacing sampling. To be specific, we use the set of frequencies such that their corresponding periods are given by

$$\tau_i = \tau_{\min} + (\tau_{\max} - \tau_{\min})^{i/d} \quad i = 1, \ldots, d.$$

Since the above argument also applies to Mercer time embedding method, we use the same initialization approach as well.

Figure 4: Training efficiency of the proposed *Bochner non-para*, the convolutional sequence embedding method *Caser* and RNN-based method on the *MovieLens* dataset. On the y-axis is the NDCG@10 on testing data.

### A.6.2 Training efficiency

The Adam optimizer is used for all models. We set the learning rate to 0.001 and set the exponential decay rate for second moment statistics to 0.98. The training is stopped if the validation metric stops increasing in 10 consecutive epochs. We use NDCG@10 for the proprietary *Walmart.com* dataset and *MovieLens* dataset, and accuracy for *Stack Overflow* data as the monitoring metric. The final metrics are computed on the hold-out test data using model checkpoints saved during training that has the best validation performances. All models are trained in TensorFlow(1.13) on a single Nvidia V100 GPU.

The training efficiency evaluations are provided in Figure 4. While it takes 9.2 seconds to train each epoch for the convolutional model *Caser* and up to 17.7 seconds for RNN-based model, it takes only 1.5 seconds for the proposed *Bochner non-para* method. Also, the test NDCG@10 reaches 0.55 within 100 seconds, while the convolutional model and RNN model reaches the same performance after 600 seconds. The training efficiency for Mercer time embedding is similar to the reported *Bochner non-para*.

### A.7 Sensitivity analysis

Figure 5: Sensitivity analysis for embedding dimensions on *MovieLens* data

We provide sensitivity analysis on time embedding dimensions for the experiments on *MovieLens* and the proprietary *Walmart.com* dataset. We focus on the *Bochner non-para* and Mercer time embedding, which we find to have the best performances. The results are plotted in Figure 5 and 6. For the recommendation outcomes on *MovieLens* dataset, we see that for both time embedding methods, the performances increase first and then stabilize as the dimension gets higher. On the proprietary dataset, the performance keeps increasing with larger time embedding dimensions. Firstly, the results suggest both time embedding methods have consistent performances on the two datasets. Secondly, we comment that the difference in data volumes might have caused the different trends

Figure 6: Sensitivity analysis for embedding dimensions on *Walmart.com* data

on the two datasets. The *Walmart.com* dataset is much larger than the *MovieLens* dataset, and the temporal and time-event interaction patterns are more complicated than that of *MovieLens*. Therefore both time embedding methods keep learning with larger time embedding dimensions.

In a nutshell, the sensitivity analysis suggests that the proposed *Bochner non-para* and Mercer time embedding give stable and consistent performances on the two datasets.

### A.8 Cases study for attention weights

In this section, we present two user-event interaction sequences sampled from the *Walmart.com* dataset and show how the attention weights progress with respect to the occurrence time of the next event (Figure 7 and 8). The sequence of user activities starts from the top to bottom. Each activity consists of the type of user behavior and the product.

Figure 7: Dynamics of attention weights on each event-action pair with respect to the next event's occurrence time, for a real-world customer online shopping sequence in home furniture.

In Figure 7, it is evident that right after the final event, actions such as *view* and *search* have high attention weights, as they reflect the most immediate interests. As for the *transaction* activity, the attention on *transaction-sofa* pair gradually rise from zero as time elapsed. The attention of *view-coffee table* pair increases over time as well. The patterns captured by our model are highly reasonable in e-commerce settings: 1. customers' short-term behaviors are more relevant to what they recently searched and viewed; 2. the long-term behaviors are affected by the actual purchases, and the products that they searched/viewed but haven't yet purchased. The attention weight dynamics reflected in Figure 8 also show similar patterns.

### A.9 Visualization of time embeddings and time kernel functions

In Figure 9, we plot the time embeddings functions $\Phi(t)$ and the corresponding kernel function $\mathcal{K}(t_1, t_2)$. Firstly, we observe that the kernel functions approximated by either Bochner time embedding or Mercer time embedding are PSD and translation-invariant (since the non-zero elements are distributed on fringes that are parallel to the diagonal in the lower panels of Figure 9). Secondly,

Figure 8: Dynamics of attention weights on each event-action pair with respect to the next event's occurrence time, for a real-world user online shopping sequence in TV and related electronics.

the visualizations show that the time embedding functions $\Phi(t)$ do capture temporal patterns, because otherwise the values in the $\Phi(t)$ matrices would be randomly distributed, as opposed to the recognizable patterns in the upper panels of Figure 9.

Figure 9: Visualization of the learned *Bochner non-para* and Mercer time embedding functions $\Phi$ (upper panel) and corresponding time kernel function $\mathcal{K}$ (lower panel). For the Mercer time embeddings, we sample three periodic kernels $\mathcal{K}_\omega$ and visualize them with their corresponding time embedding functions.

## A.10  Reference implementation

The reference code for our implementations is provided in the supplementary material.

## Footnotes

[5]https://www.tensorflow.org/api_docs/python/tf/contrib/distributions/bijectors/MaskedAutoregressiveFlow

[6]https://www.tensorflow.org/api_docs/python/tf/contrib/distributions/bijectors/RealNVP


[Supplementary Material 2]

# SELF-ATTENTION WITH FUNCTIONAL TIME REPRESENTATION LEARNING

Da Xu*, Chuanwei Ruan*, Sushant Kumar, Evren Korpeoglu, Kannan Achan

Walmart Labs

## Self-attention for Continuous-time Sequence Modelling?

- Self-attention mechanism [4] is powerful but only works on discrete-time sequence with positional encoding.
- Time spans between sequential events often carry important signals.
- We identify the forms of functional time mapping that work well with self-attention especially the scaled dot-product attention.
- The proposed approaches have solid theoretical justification and guarantees. Experiments demonstrate its great practical values on real-world continuous-time sequence datasets.

## Preliminaries in Functional Analysis

**Temporal kernel.** Embedding time from an interval $T = [0, t_{\max}]$ to $\mathbb{R}^d$ is equivalent to finding a mapping $\Phi: T \to \mathbb{R}^d$. Due to the **inner product** formulation of self-attention [4] and the **translation invariant** property of time spans, we define the *temporal kernel* as $\mathcal{K}: T \times T \to \mathbb{R}$ where $\mathcal{K}(t_1, t_2) := \langle \Phi(t_1), \Phi(t_2) \rangle$ and $\mathcal{K}(t_1, t_2) = \psi(t_1 - t_2), \forall t_1, t_2 \in T$ for some $\psi: [-t_{\max}, t_{\max}] \to \mathbb{R}$.

**Embedding as feature maps.** The feature map $\Phi$ captures how the temporal kernel function embeds the original time data into a higher dimensional space. So the task of learning temporal patterns is converted to a kernel learning problem with $\Phi$ as feature map.

**Theorem 1 (Bochner's Theorem [1]).** *A continuous, translation-invariant kernel $\mathcal{K}(\mathbf{x}, \mathbf{y}) = \psi(\mathbf{x} - \mathbf{y})$ on $\mathbb{R}^d$ is positive definite if and only if there exists a non-negative measure on $\mathbb{R}$ such that $\psi$ is the Fourier transform of the measure.*

**Implications:** when scaled properly we can express $\mathcal{K}$ with:

$$\mathcal{K}(t_1, t_2) = \psi(t_1, t_2) = \int_{\mathbb{R}} e^{i\omega(t_1 - t_2)} p(\omega) d\omega = E_\omega[\xi_\omega(t_1) \xi_\omega(t_2)^*], \quad (1)$$

where $\xi_\omega(t) = e^{i\omega t}$. Since the kernel $\mathcal{K}$ and the probability measure $p(\omega)$ are real, we extract the real part of (1) and obtain an alternate expression of the kernel:

$$\mathcal{K}(t_1, t_2) = E_\omega\big[\cos(\omega(t_1 - t_2))\big] = E_\omega\big[\cos(\omega t_1)\cos(\omega t_2) + \sin(\omega t_1)\sin(\omega t_2)\big]. \quad (2)$$

**Theorem 2 (Mercer Theorem [2]).** *Consider the function class $L^2(\mathcal{X}, \mathbb{P})$ where $\mathcal{X}$ is compact. Suppose that the kernel function $\mathcal{K}$ is continuous with positive semidefinite and satisfy the condition $\int_{\mathcal{X} \times \mathcal{X}} \mathcal{K}^2(x, z) d\mathbb{P}(x) d\mathbb{P}(y) \leq \infty$, then there exist a sequence of eigenfunctions $(\phi_i)_{i=1}^\infty$ that form an orthonormal basis of $L^2(\mathcal{X}, \mathbb{P})$, and an associated set of non-negative eigenvalues $(c_i)_{i=1}^\infty$ such that*

$$\mathcal{K}(x, z) = \sum_{i=1}^\infty c_i \phi_i(x) \phi_i(z), \quad (3)$$

*where the convergence of the infinite series holds absolutely and uniformly.*

**Implications:** we can embed instances from the functional time domain $T$ into the infinite sequence space $\ell^2(\mathbb{N})$, by defining the mapping via $t \mapsto \Phi^{\mathcal{M}}(t) := [\sqrt{c_1}\phi_1(t), \sqrt{c_2}\phi_2(t), \dots]$, and Mercer's Theorem guarantees the convergence of $\langle \Phi^{\mathcal{M}}(t_1), \Phi^{\mathcal{M}}(t_2) \rangle \to \mathcal{K}(t_1, t_2)$.

**Note.** We still haven't reached a feasible parametric form for $\Phi$, since the $p(\omega)$ in Bochner's and the set of basis $\{\phi_i\}$ in Mercer's are unknown.

## Proposed Approaches

**Bochner's encoding.** Following the implications of *Bochner's* Theorem, the expectation in (1) can be approximated by Monte Carlo integral [3]. With $d$ samples drawn from $p(\omega)$, an estimate of our kernel $\mathcal{K}(t_1, t_2)$ can be constructed by $\frac{1}{d}\sum_{i=1}^d \cos(\omega_i t_1)\cos(\omega_i t_2) + \sin(\omega_i t_1)\sin(\omega_i t_2)$. So we propose finite dimensional Bochner feature map:

$$t \mapsto \Phi_d^{\mathcal{B}}(t) := \sqrt{\frac{1}{d}}\big[\cos(\omega_1 t), \sin(\omega_1 t), \dots, \cos(\omega_d t), \sin(\omega_d t)\big],$$

and we prove the following claim which guarantees the stochastic uniform convergence.

**Claim 1.** *Let $p(\omega)$ be the corresponding probability measure stated in Bochner's Theorem for kernel function $\mathcal{K}$. Suppose the feature map $\Phi$ is constructed as described above using samples $\{\omega_i\}_{i=1}^d$, we have*

$$Pr\Big(\sup_{t_1, t_2 \in T} \big|\Phi_d^{\mathcal{B}}(t_1)'\Phi_d^{\mathcal{B}}(t_2) - \mathcal{K}(t_1, t_2)\big| \geq \epsilon\Big) \leq 4\sigma_p \sqrt{\frac{t_{\max}}{\epsilon}} exp\Big(\frac{-d\epsilon^2}{32}\Big), \quad (4)$$

*where $\sigma_p^2$ is the second momentum with respect to $p(\omega)$.*

Therefore, we can either use parametric or non-parametric distributional learning methods to obtain samples from the optimized $p(\omega)$, and then construct $\Phi_d^{\mathcal{B}}$ accordingly.

**Mercer's encoding.** As for the *Mercer* Theorem, we prove in the following Proposition 1 that a straightforward parameterization of the feature map via the Fourier basis expansion is possible, by decomposing the temporal kernel $\mathcal{K}$ into a set of periodic kernel functions $\{\mathcal{K}_\omega\}$.

**Proposition 1.** *For kernel function $\mathcal{K}$ that is continuous, PSD and translation-invariant with $\mathcal{K} = \psi(t_1 - t_2)$, suppose $\psi$ is a even periodic function with frequency $\omega$, i.e $\psi(t) = \psi(-t)$ and $\psi\big(t + \frac{2k}{\omega}\big) = \psi(t)$ for all $t \in [-\frac{1}{\omega}, \frac{1}{\omega}]$ and integers $k \in \mathbb{Z}$, the eigenfunctions of $\mathcal{K}$ are given by the Fourier basis.*

After truncating the series of Fourier basis, we have the infinite dimensional Mercer's feature map for each $\mathcal{K}_\omega$:

$$t \mapsto \Phi_\omega^{\mathcal{M}}(t) = \Big[\sqrt{c_1}, \dots, \sqrt{c_{2j}}\cos\big(\frac{j\pi t}{\omega}\big), \sqrt{c_{2j+1}}\sin\big(\frac{j\pi t}{\omega}\big), \dots\Big],$$

where $c_j$ are the corresponding Fourier coefficients and $\omega_i$ are free model parameters.

Fig. 1: Left panel: visual illustration of the proposed Bochner and Mercer time embedding ($\Phi_d^{\mathcal{B}}(t)$ and $\Phi_{\omega,d}^{\mathcal{M}}(t)$) for a specific $t = t_i$ with $d = 3$. Right panel: network architecture for next-event prediction at time $t$ with a single block.

## Experiments and Results

**Time-event interaction:** we first concatenate the event embedding and time representations into $[\mathbf{Z}, \mathbf{Z}_T]$ where $\mathbf{Z} = [Z_1, \dots, Z_q]$, $\mathbf{Z}_T = [\Phi(t_1), \dots, \Phi(t_q)]$ and then project them into the $\mathbf{Q}$, $\mathbf{K}$ and $\mathbf{V}$ spaces respectively to capture their linear or non-linear interactions, e.g.

$$\mathbf{Q} = \text{ReLU}([\mathbf{Z}, \mathbf{Z}_T]\mathbf{W}_0 + b_0)\mathbf{W}_1 + b_1,$$

and finally we use $\mathbf{h}^{(i)} = \text{Attn}^{(i)}(\mathbf{Q}, \mathbf{K}, \mathbf{V})$ as the hidden output of the $i^{th}$ head in the multi-head attention setting.

| Feature maps specified by $[\phi_{2i}(t), \phi_{2i+1}(t)]$ | Origin | Parameters | Interpretations of $\omega$ |
|---|---|---|---|
| $\big[\cos(\omega_i(\mu)t), \sin(\omega_i(\mu)t)\big]$ | Bochner Normal | $\mu$: location-scale parameters specified for the *reparametrization trick*. | $\omega_i(\mu)$: converts the $i^{th}$ sample (drawn from auxiliary distribution) to target distribution under location-scale parameter $\mu$. |
| $\big[\cos(g_\theta(\omega_i)t), \sin(g_\theta(\omega_i)t)\big]$ | Bochner Inv CDF | $\theta$: parameters for the inverse CDF $F = g_\theta$. | $\omega_i$: the $i^{th}$ sample drawn from the auxiliary distribution. |
| $[\cos(\tilde{\omega}_i t), \sin(\tilde{\omega}_i t)]$ | Bochner Non-param | $\{\tilde{\omega}\}_{i=1}^d$: transformed samples under non-parametric inverse CDF transformation. | $\tilde{\omega}_i$: the $i^{th}$ sample of the underlying distribution $p(\omega)$ in Bochner's Theorem. |
| $\big[\sqrt{c_{2i,k}}\cos(\omega_j t), \sqrt{c_{2i+1,k}}\sin(\omega_j t)\big]$ | Mercer | $\{c_{i,k}\}_{i=1}^{2d}$: the Fourier coefficients of corresponding $\mathcal{K}_{\omega_j}$, for $j = 1, \dots, k$. | $\omega_j$: the frequency for kernel function $\mathcal{K}_{\omega_j}$ (can be parameters). |

Fig. 2: A summary of the proposed approaches.

We compare among the proposed function mapping methods (the details are shown in Figure 2) in self-attention and with state-of-the-art baseline approaches including self-attention with positional encoding (PosEnc), on recommendation tasks with the **Stack Overflow**, **Moivelens** and **Walmart.com** datasets. Case studies and analysis on the attention weights are given in Fig. 3. The results are provided in Fig. 4.

Fig. 3: Attention weight analysis as functions of time and event.

| Stack Overflow | | | | | | | | |
|---|---|---|---|---|---|---|---|---|
| **Method** | LSTM | TimeJoint | RMTPP | PosEnc | **Bochner Normal** | **Bochner Inv CDF** | **Bochner Non-para** | **Mercer** |
| **Accuracy** | 46.03(.21) | 46.30(.23) | 46.23(.24) | 44.03(.33) | 44.89(.46) | 44.67(.38) | 46.27(0.29) | **46.83(0.20)** |
| config | | | | | | NVP | | $k = 10$ |

| MovieLens-1m | | | | | | | | |
|---|---|---|---|---|---|---|---|---|
| **Method** | GRU4Rec | Caser | TransRec | | | | | |
| **Hit@10** | 75.01(.25) | 78.86(.22) | 64.15(.27) | 82.45(.31) | 81.60(.69) | 82.52(.36) | 82.86(.22) | **82.92(.17)** |
| **NDCG@10** | 55.13(.14) | 55.38(.15) | 39.72(.16) | 59.05(.14) | 59.47(.56) | 60.80(.47) | 60.83(.15) | **60.88(.11)** |
| config | | | | | | MAF | | $k = 5$ |

| Walmart.com data | | | | | | | | |
|---|---|---|---|---|---|---|---|---|
| **Method** | GRU4Rec | RNN+attn | TransRec | | | | | |
| **Hit@5** | 4.12(.19) | 5.90(.17) | 7.03(.15) | 8.63(.16) | 4.27(.91) | 9.04(.31) | 9.25(.15) | **10.92(.13)** |
| **NDCG@5** | 4.03(.20) | 4.66(.17) | 5.62(.17) | 6.92(.16) | 4.06(.94) | 7.27(.26) | 7.34(.12) | **8.90(.11)** |
| **Hit@10** | 6.71(.50) | 9.03(.44) | 10.38(.41) | 12.49(.38) | 7.66(.92) | 12.77(.65) | 13.16(.41) | **14.94(.31)** |
| **NDCG@10** | 4.97(.31) | 7.36(.26) | 8.72(.26) | 10.84(.26) | 6.02(.99) | 10.95(.74) | 11.36(.27) | **12.81(.22)** |
| config | | | | | | MAF | | $k = 25$ |

Fig. 4: Performance metrics for the proposed approach and baseline models. MAF and NVP are the flow-based distribution learning methods, and $k$ gives the dimension of Fourier basis expansions.

[1] Lynn H Loomis. *Introduction to abstract harmonic analysis*. Courier Corporation, 2013.

[2] James Mercer. "Xvi. functions of positive and negative type, and their connection the theory of integral equations". In: *Philosophical transactions of the royal society of London. Series A, containing papers of a mathematical or physical character* 209.441-458 (1909), pp. 415–446.

[3] Ali Rahimi and Benjamin Recht. "Random features for large-scale kernel machines". In: *Advances in neural information processing systems*. 2008, pp. 1177–1184.

[4] Ashish Vaswani et al. "Attention is all you need". In: *Advances in neural information processing systems*. 2017, pp. 5998–6008.