[Reviews · NeurIPS 2019]

Reviewer 1



Originality: The application of self-attention in continuous-time event sequences is an interesting approach. The authors clearly note the shortcoming of self-attention when applied to such problems. They propose translation-invariant time kernel functions justified by classic function analysis theories and implement 4 new time embeddings that can be optimized by backpropagation and are compatible with self-attention. I believe the proposed time embeddings are novel and generalizable to other temporal tasks. Quality: Motivation from classic functional analysis theory [12] and [14]and developing differentiable time embeddings is the key contribution of this paper. The derivation of these embeddings from the original theorems is very clear and impressive and is the most interesting contribution of this paper. However, the lack of evidence showing the advantage of these embeddings in the results (as I explain in significance) lowers the quality of overall paper. Clarity: The paper is well-written clearly stating the shortcomings of the previous works and how the proposed approach takes into account those shortcomings. There are few typos which can be corrected e.g line 244 (supplemnetary meterial). It’s not clear in Figure 1 what the items in legend represents and therefore its hard to interpret the results in Figure 1. Significance: Although the proposed approach is novel, the results do not convince me about its significance. For instance, none of the 4 proposed time embeddings (Table 1) significantly outperform the position encoding in all 3 datasets. The proposed Bochner Normal and Bochner Inv CDF either perform similarly or worse than position encodings. While Mercer shows the best results in 2 datasets it fails to outperform in movielens dataset. Hence, the advantage over position encoding doesn’t seem evident from the results.

Reviewer 2



UPDATE: I have read and thank the authors for their response to my original questions. ----------------- I enjoyed reading this paper, thanks! Some detailed feedback: L17: "capture"->"capturing" L88: "Embed" -> "Embedding"; "find" -> "finding" * Use ` and ' for single quotes, eg L147. L219 "a"->"an"

Reviewer 3



Originality: Positional/temporal embeddings with self-attention models have been studied in many papers, although not with theoretical-driven motivations. The paper should distinguish the contributions more clearly. Quality: The quality of theory-driven discussions is high but the experimental results section needs to be improved. Significance: The paper addresses an important direction to enable self-attention models for time-series forecasting. In general, the study of temporal embedding selection is impactful, especially with the insights from two theorems. Clarity: The paper is mostly well-written and the contributions are clearly explained. But I think there is room for improvement in writing. For example, some theoretical arguments can be moved to the supplementary material, in exchange for interesting interpretations and analysis moved to the main body of the paper.

[Author Response · NeurIPS 2019]

We'd like to thank the reviewers for their careful reading and valuable comments. First, we want to emphasize that the novelty of the proposed method, which addresses how to embed continuous time to differentiable functional domain that works with self-attention, comes with substantial theoretical derivations and superior practical performances. We believe this is a useful contribution for both functional representation learning and deep temporal sequence learning.

Second, we apologize for typos, grammar mistakes and unclear notations. They will be corrected in the final version. In Fig 1a (1c), the *SEARCH/ATC/VIEW/TRX* in legend stands for user actions of *search/add to cart/view/purchase* in online shopping. It shows that by combining time and event representations, the model captures useful time-event interactions, e.g. the products searched by a user gets higher attention weight when recommending the next product.

Third, we provide additional experiment results in Table 1. Previously, *Mercer* time embedding uses $k = 30$ as the degree for Fourier basis under each frequency in all experiments. We now treat $k$ as a tuning parameter and report the best performances. For the *Bochner Inv CDF* method, we employ two SOTA flow-based inverse CDF learning methods, i.e. masked autoregressive flow (*MAF*) and non-volume preserving (*NVP*) transformation, in addition to *MLP*.

| Dataset | Stack Overflow | Movielens | | eComerce | |
|---|---|---|---|---|---|
| Metric | Accuracy | Hit@10 | NDCG@10 | HIT@10 | NDCG@10 |
| **Mercer** | **46.53(.20), [k=10]** | **82.92(.16)** | **60.88(.11), [k=5]** | **14.94(.31)** | **12.81(.22), [k=25]** |
| **Bochner** | 40.47(.65), [*MLP*] | 81.60(.65) | 60.60(.53), [*MLP*] | 9.84(.86) | 7.95(.94), [*MLP*] |
| **Inv CDF** | 42.01(.39), [*MAF*] | *82.52(.36)* | *60.80(.47), [MAF]* | *12.77(.65)* | *10.95(.74), [MAF]* |
| | *42.13(.38), [NVP]* | 82.37(.38) | 60.55(.50), [*NVP*] | 12.38(.68) | 10.62(.73), [*NVP*] |
| **PosEnc** | 41.09(.33) | 82.45(.31) | 59.05(.14) | 12.49(.38) | 10.84(.26) |

Table 1: Additional experiment results (converted to percentage by multiplying by 100). The model configurations are reported in the square brackets.

Figure 1: Sensitivity analysis on degree $k$ under different time embedding dimensions.

**To reviewer #1. Q1: Why Mercer fails to outperform in Movielens dataset?** With the tuned Fourier basis degree $k$, Mercer's method consistently outperforms others across all tasks. While $d$, the dimension of time embedding, controls how well the bandwidth of $[\omega_{\min}, \omega_{\max}]$ is covered, $k$ controls the degree of freedom for the Fourier basis under each frequency. When $d$ is fixed, larger $k$ may lead to overfitting issue for the time kernels under certain frequencies, which is confirmed by the sensitivity analysis on $k$ provided in Figure 1 for the Movielens dataset.

**Q2: The insight/explanation/analysis of why other 3 methods do not perform well.** After employing the SOTA inverse CDF learning methods, *Bochner Inv CDF* achieves better performances than positional encoding and other baselines on Movlielens and eCommerce dataset. This suggests the importance of having higher model complexity for learning $p(\omega)$ under Bochner's Thm, which also explains why *Bochner Normal* fails in most cases since normal distribution has limited capacity in capturing complicated distributional signals. On the other hand, *Bochner Non-para* is actually the special case of *Mercer*'s method with $k = 1$ and no intercept. While Bochner's methods originate from random feature sampling, Mercer's method grounds in functional basis expansion. In practice, we may expect Mercer's method to give more stable performances since it does not rely on distributional learning and sampling. However, with advancements in Bayesian deep learning and probabilistic computation, we may also expect *Bochner Inv CDF* to work well with proper distribution learning models (as we have shown above with the flow-based methods).

**To reviewer #2. Q1: Connections to other tasks that would make the proposed approach more generally applicable?** Besides recommender systems, temporal sequence learning has wide applications in clinical trial analysis, temporal network representation learning and reinforcement learning. The idea of functional representation learning, on the other hand, is not restricted to temporal sequence learning. The use of positional encoding with self-attention in NLP tasks requires a fixed input sequence length and a large number of parameters when sentences are long. The proposed approach can be easily adapted to functional position encoding, which does not suffer from the above drawbacks.

**To reviewer #3. Q1: Relations to related position/temporal encoding methods?** To the best of our knowledge, the other positional/temporal encoding methods either do not handle continuous time or are driven by heuristics (we will include the missing references). The original fixed positional encoding also takes the form of sinusoidal functions and can be thought of as a special version of *Bochner Non-para* method. While their frequencies are mostly chosen by insights, in our approach, the frequencies are either free model parameters or sampled from learnable distributions.

**Q2: Motivation for capturing the relationships in dot product.** The motivation mainly comes from the key-query inner product formulation of the self-attention model in (1).

**Q3-Q7:** In the additional experiments, the flow-based inverse CDF learning methods give much better performances than the three-layer *MLP*, which was originally chosen for illustration purposes. Given the limited space here, the detailed implementation for comparisons models will be described in full in the final version. An algorithm description (flow chart) will also be added to the appendix. We will address all remaining suggestions in the final version.

[Meta-Review · NeurIPS 2019]

The main contribution of the work is a novel approach to embed continuous time to differentiable functional domain in such a way that is compatible with modern models using self-attention. All reviewers agree that the derivation of these embeddings from function analysis results is quite interesting and contrasts with the intuitive derivations presented in the literature. In their response, the authors included improved experimental results by tuning a hyper parameter that was previously set to a fixed value (the degree for Fourier basis under each frequency in the Mercer embedding). Reviewers 1 and 3 still point out that the experimental section could be stronger. While the AC agrees that more experiments would make the paper stronger, the paper contains several interesting ideas and is able to show enough empirical evidence that the method has practical relevance. The AC encourages the authors to follow the recommendation of reviewer 2 and include the missing references as well as some insights of why the 3 of the proposed methods do not perform well (in the lines of what was included in the response).